# Advances in Therapeutics to Alleviate Cognitive Decline and Neuropsychiatric Symptoms of Alzheimer’s Disease

**DOI:** 10.3390/ijms25105169

**Published:** 2024-05-09

**Authors:** Jialin Li, Anita Haj Ebrahimi, Afia B. Ali

**Affiliations:** UCL School of Pharmacy, University College London, 29-39 Brunswick Square, London WC1N 1AX, UK; jialin_li.20@ucl.ac.uk (J.L.); anita.ebrahimi.19@alumni.ucl.ac.uk (A.H.E.)

**Keywords:** Alzheimer’s disease, anxiety disorder, depressive disorder, GABA_A_ receptors (GABA_A_Rs), benzodiazepines, selective serotonin reuptake inhibitors (SSRIs), selective noradrenaline reuptake inhibitors (SNRIs), neuroinflammation, microglial, astrocytes, FDA-approved drugs, endothelin B receptor (ET_B_R), repurposing drugs

## Abstract

Dementia exists as a ‘progressive clinical syndrome of deteriorating mental function significant enough to interfere with activities of daily living’, with the most prevalent type of dementia being Alzheimer’s disease (AD), accounting for about 80% of diagnosed cases. AD is associated with an increased risk of comorbidity with other clinical conditions such as hypertension, diabetes, and neuropsychiatric symptoms (NPS) including, agitation, anxiety, and depression as well as increased mortality in late life. For example, up to 70% of patients diagnosed with AD are affected by anxiety. As aging is the major risk factor for AD, this represents a huge global burden in ageing populations. Over the last 10 years, significant efforts have been made to recognize the complexity of AD and understand the aetiology and pathophysiology of the disease as well as biomarkers for early detection. Yet, earlier treatment options, including acetylcholinesterase inhibitors and glutamate receptor regulators, have been limited as they work by targeting the symptoms, with only the more recent FDA-approved drugs being designed to target amyloid-β protein with the aim of slowing down the progression of the disease. However, these drugs may only help temporarily, cannot stop or reverse the disease, and do not act by reducing NPS associated with AD. The first-line treatment options for the management of NPS are selective serotonin reuptake inhibitors/selective noradrenaline reuptake inhibitors (SSRIs/SNRIs) targeting the monoaminergic system; however, they are not rational drug choices for the management of anxiety disorders since the GABAergic system has a prominent role in their development. Considering the overall treatment failures and side effects of currently available medication, there is an unmet clinical need for rationally designed therapies for anxiety disorders associated with AD. In this review, we summarize the current status of the therapy of AD and aim to highlight novel angles for future drug therapy in our ongoing efforts to alleviate the cognitive deficits and NPS associated with this devastating disease.

## 1. Introduction

### General Overview of the Pathology of Alzheimer’s Disease

Dementia is one of the most common neurological disorders affecting memory and behaviour. Alzheimer’s disease (AD) is the most common form of dementia, constituting a minimum of 60% of dementia diagnoses [1,2,3].

The symptoms associated with AD include specific onset and progression of cognitive and functional deterioration, such as neuronal loss and cognitive deficits [4,5]. AD is one of the leading mortality factors among the elderly. At least 50 million people worldwide are diagnosed with AD, and the rate is expected to rise to 152 million by 2050 [6]. Over the previous decade in the United Kingdom (UK), the number of deaths due to AD tripled from 4.23% to 12.53%. In 2019, AD was the fifth most common cause of mortality among those above 64 years old [7], believed to have a median survival duration of 8–10 years from diagnosis, with some exceptions where patients survived >20 years. The major risk factors for this disease include genetics; environmental factors; age; gender; lifestyle choices; comorbidities that influence disease onset, including vascular disease, diabetes, and infection; and traumatic brain injury (TBI). Familial AD is a hereditary form of AD caused by mutations in certain genes; notably, a mutation due to a polymorphism on chromosome 19 forms ApoE4 lipoprotein, which is inefficient at removing amyloid beta (Aβ) plaques from the brain’s cortical regions, thereby escalating AD risk, and presenilin 1 and 2 genes, responsible for the catalytic subunit of γ-secretase transcription, which have been reviewed previously [8,9,10].

According to the World Health Organization (WHO, 2021), neuropsychiatric symptoms (NPS) including anxiety and depression are the leading cause of disability and a high rate of inadequate treatment is a major concern [11,12]. The impact of the COVID-19 pandemic has increased the prevalence of depression and anxiety by 27.88% and 26.35%, respectively, in comparison to the pre-pandemic era in the UK, which indicates the growing need for effective interventions [13]. Furthermore, anxiety disorders are among the most prevalent and disabling NPS disorders worldwide; in the UK alone, one in six adults per given week are affected by NPS, with ~10% of the UK general population taking anxiolytics [14]. However, up to 70% of patients diagnosed with AD are affected by anxiety [15,16]. As ageing is the major risk factor for AD, this represents a huge global burden in ageing populations.

## 2. Clinical Manifestations and Conventional Therapeutic Approaches for AD to Alleviate Cognitive Decline

Patients with AD typically experience linguistic impairment, poor judgement, disruptive behaviour, increased memory loss, disorientation, and difficulty learning new things [7,17]. However, the diagnostic procedure for AD is complex as it requires the exclusion of conditions that can have similar symptoms, such as infection and depression, and there is no single test. Radiologically, images from computed tomography (CT) scans, magnetic resonance imaging (MRI), and positron emission tomography (PET) scans [18] can provide an indication of brain atrophy and amyloid plaques which are associated with the disease [19,20,21]. Histologically, the abundant extracellular accumulations of amyloid plaques and intracellular neurofibrillary tangles (NFTs) are the main hallmarks of AD, which may interfere with the normal communication between neurons and disrupt the brain’s ability to function properly [4,22,23,24]. Studies have indicated a correlation between the severity of the cognitive deficits associated with AD and hyperaccumulation of plaques and NFTs [25].

Conventional treatments for AD primarily focus on symptom management and enhancing the quality of life of patients. Until now, AD patients had limited treatment choices, primarily consisting of supportive measures such as cognitive stimulation therapy (CST) or reminiscence. In addition to these, non-pharmacological approaches including cognitive stimulation, physical exercise, and a balanced diet also play an important role. Occupational therapy can also provide significant assistance in preserving patient independence and managing their daily activities.

Furthermore, recent research indicates the significant potential of combination therapies for AD, which can be extremely beneficial in slowing the progression of the disease and improving quality of life [26,27]. Such treatments typically involve a combination of medications, lifestyle changes, and non-pharmacological interventions such as cognitive stimulation, physical exercise, and dietary changes [28]. Notably, the application of combination therapies has demonstrated greater efficacy compared to single treatments, offering notable improvements in cognitive function and slowing disease progression.

### Current FDA-Approved Treatments for AD-Associated Cognitive Deficits

Current therapeutics and novel drug strategies have always been trends in discussion to further our understanding and design of newer drug therapies for dementia. In general, recent reviews also cover these topical themes [29,30,31]. Current treatments for AD primarily aim to alleviate symptoms, with strategies ranging from patient care and support to exercise programs, which are associated with a reduced risk of dementia [32]. Pharmacological treatments include antidepressants, antipsychotics, and notably, acetylcholinesterase inhibitors, a widely used approach [33]. Despite their effectiveness in symptom management, these treatments do not address the underlying causes of AD. Moreover, they can lead to undesirable side effects like vomiting and headaches or hallucinations, highlighting a significant gap in pharmacological strategies for treating the disease.

Until 2020, only four FDA-approved drugs for AD were shown to provide a modest benefit in symptom management but not to halt disease progression: donepezil, galantamine, rivastigmine, and memantine (the first three are inhibitors of the acetylcholinesterase (AChE) enzyme [17] and the latter is a glutamate N-methyl-D-aspartate (NMDA) receptor antagonist [19]). AChE inhibitors are used to compensate for the low levels of acetylcholine (ACh) in the brain in AD. They work by limiting the breakdown of ACh into harmful products, not by treating the underlying cause that leads to the increased breakdown, and have shown moderate efficacy and safety in patients with moderate-to-severe AD [34]. While AChE inhibitors are usually well tolerated by patients, they may also produce significant gastrointestinal side effects. However, memantine has a different mechanism of action than cholinergic drugs and is thought to be neuroprotective [35]. Memantine is recommended for more severe cases of dementia [36] and is also an agonist for certain dopamine receptors as well as an antagonist for NMDA receptors, which are part of excitatory glutamatergic transmission. Seeing as it is thought that glutamatergic transmission is impaired in AD (hyperactive) and causes neuronal excitotoxicity, the use of memantine as a low-affinity glutamate/NMDA receptor blocker reduces the excess Ca^2+^ transmission and lowers excitotoxicity. However, memantine, like AChEs, treats symptoms rather than actually prevent disease progression, and most patients do not show any real benefit in the long term [35,37]. This situation shows the urgent need for more effective treatments that directly target the main causes of AD.

Excitingly, over the past two years, antibody therapy has emerged as a breakthrough in AD treatment, shown in Table 1. Aduhelm (aducanumab), the fifth FDA-approved drug for AD, is an example of this therapy and received approval in the United States in June 2021 [22,38,39,40]. Further progressing antibody therapies, Leqembi (lecanemab-irmb) was approved in January 2023 for the treatment of early-stage AD [41,42]. Both treatments are monoclonal antibodies, meaning that they are designed to bind to a specific protein (in this case, amyloid-β) to reduce its levels in the brain and slow the progression of the disease [39,42]. Aducanumab, as the first drug to remove amyloid-β, effectively alleviates the cognitive and functional impairments caused by AD, especially for people living with early AD [43]. After a controversial accelerated FDA approval, Biogen has decided to halt the development and commercialization of aducanumab due to little evidence proving that reduction of amyloid helped patients improve their memory and cognitive problems. Additionally, trials indicated potential risks of brain swelling and bleeding associated with the drug [44,45]. Instead, Biogen will shift its focus to developing lecanemab [46]. Lecanemab is also used to treat AD patients with mild cognitive impairment or mild dementia with a known amyloid-β pathology [47]. This makes antibody therapy the most promising and groundbreaking treatment for AD. It preferentially targets soluble aggregated Aβ and works on Aβ oligomers, protofibrils, and insoluble fibrils [48,49]. Lecanemab was found to reduce the progression of AD by about 27% [50].

Recently, Eli Lilly reported a new AD drug ‘Donanemab (LY3002813)’, which is also an antibody therapy [51,52]. Similar to the previous antibody therapies, donanemab is still a humanized IgG1 monoclonal antibody, designed to treat early AD by targeting a different form of Aβ to clear the existing Aβ plaques to slow the decline in cognitive function associated with AD. Donanemab is targeted against an epitope at the N-terminal of a specific type of Aβ—pyroglutamate Aβ—which is found only in the brain amyloid plaques associated with AD [52]. A phase II study conducted by Eli Lilly reported that donanemab slowed the progression of AD by about 35%, showing greater ability in clearing amyloid plaques compared to previous drugs [51]. However, antibody therapies have only been tested on people with early stages of the disease and it is unknown how effective they are for more advanced forms.

Despite antibody therapies showing promise in alleviating symptoms of AD, they still face several challenges that limit their efficacy and applicability. Primarily, these therapies are used to treat early-stage AD patients and can only slow down but not halt the disease progression [53]. Moreover, current antibody treatments are only targeted to Aβ; however, the pathology of AD is complex. Beyond Aβ accumulation, other critical aspects of AD pathology, such as NFTs (tau) and neuroinflammation, are increasingly being recognized as important factors in the disease’s progression. All these imply that we need to find new treatment options for AD. Additionally, a notable concern associated with amyloid-targeting antibody therapies is amyloid-related imaging abnormalities (ARIA) [54,55]. Aducanumab has been linked to ARIA, especially in APOE ε4 carriers, leading to a dose-dependent increase in occurrences of vasogenic edema [55]. Together, these factors suggest an urgent need to develop new AD treatment options that encompass a wider range of the disease’s complex pathology.

In addition to therapies targeting Aβ, several treatment strategies focus on tau pathology, including tau antisense oligonucleotides and anti-tau oligomer antibodies. Increasing evidence supports the role of hyperphosphorylated tau aggregation as a central contributor to neurodegeneration in AD [56,57,58]. The tau protein, primarily expressed in neurons, is encoded by the microtubule-associated protein tau (MAPT) gene [59,60]. Preclinical studies suggest that reducing tau can prevent certain deficits mediated by amyloid-β (Aβ), underscoring tau’s pivotal role in Aβ toxicity during the early stages of AD pathogenesis [61,62]. At the close of 2023, Biogen revealed new Phase 1b clinical data for BIIB080, an investigational antisense oligonucleotide (ASO) therapy targeting tau in patients with mild AD [62]. BIIB080 is engineered to target MAPT mRNA to lower tau protein production. Inhibiting MAPT expression to reduce tau levels is a crucial strategy which directly targets a key mechanism of disease that affects AD patients [61]. While BIIB080 targets the reduction of tau protein production by inhibiting MAPT mRNA, another promising strategy involves the anti-tau oligomer antibody APNmAb005 [63,64,65]. APNmAb005 is a humanized monoclonal antibody designed to block the synaptic toxicity caused by tau oligomers [65]. APNmAb005 selectively binds to tau oligomers and aggregates, primarily within the synapses of pathological brain tissues, effectively inhibiting tau propagation [64,65]. Currently, APNmAb005 is undergoing Phase 1 clinical trials to evaluate its safety and tolerability, representing a novel approach in tau immunotherapy for neurodegenerative disorders [64].

In 2024, some clinical trials are actively exploring therapies for AD. A significant trial involves AL002, targeting early AD stages and assessing changes in cognitive and biomarker outcomes over up to 96 weeks. More information is mentioned in Table 2 [66]. These trials reflect a broad effort to target various aspects of AD pathology and progression.

**Table 1 ijms-25-05169-t001:** Current, key FDA-approved treatments for AD.

Name of Drug	Class of Drug	Chemical Structure	Action of the Drug	Side Effects
Donepezil Approved in 1996 [67,68]	Acetylcholinesterase (AChE) inhibitor used to manage mild to severe symptoms of AD.	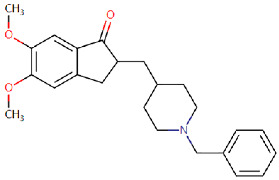	By inhibiting AChE, donepezil improves the cognitive and behavioural signs and symptoms of AD, which may include apathy, aggression, confusion, and psychosis.	Gastrointestinal side effects including nausea, vomiting, anorexia, and diarrhoea. Nervous side effects including dizziness, confusion, and insomnia.
Rivastigmine Approved in 1997 [69,70]	AChE inhibitor used to treat mild to moderate symptoms of AD.	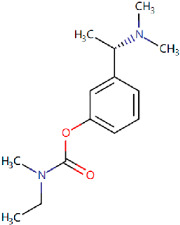	Rivastigmine inhibits both butyrylcholinesterase and AChE, preventing the hydrolysis of acetylcholine and thus leading to an increased concentration of acetylcholine at cholinergic synapses.	Gastrointestinal side effects including nausea and vomiting, decreased appetite, diarrhoea, and abdominal pain. Nervous side effects including pain, headache, dizziness, syncope, fatigue, and malaise.
Galantamine Approved in 2001 [71,72]	AChE inhibitor used to manage mild to moderate AD.	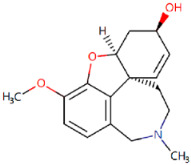	Galantamine inhibits AChE in the synaptic cleft, thereby enhancing cholinergic neuron function and signalling.	Gastrointestinal side effects including nausea, vomiting, anorexia, and abdominal pain. Nervous side effects including dizziness, pain, somnolence, and agitation.
Memantine Approved in 2013 [73,74]	NMDA receptor antagonist used to treat moderate to severe dementia in AD.	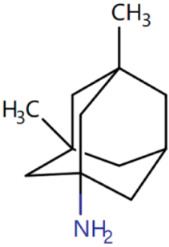	Memantine blocks the neurotransmitter glutamate from acting on NMDA receptors that are partly responsible for neuronal excitability, thus preventing hyperexcitability seen in early and late AD.	Gastrointestinal side effects including nausea and vomiting. Nervous side effects including dizziness, headache, insomnia, and confusion. Others including falls and hypertension.
Aducanumab Approved in 2021 [75,76]	Monoclonal IgG1 antibody used to treat mild symptoms of AD.	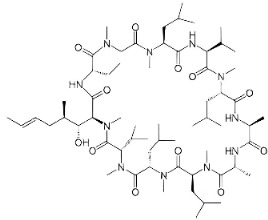	The drug binds to amyloid-β, reducing amyloid plaques in the brain. The treatment is associated with slowing the rate of progression of AD.	Nervous side effects including headache, dizziness, nausea, and confusion. Others including falls, microhaemorrhages (ARIA-H microhaemorrhage), ARIA-superficial siderosis, and generalized tonic–clonic seizures.
Lecanemab Approved in 2023 [46]	Monoclonal IgG1 antibody used to treat mild symptoms of AD.	Not available	The drug reduces Aβ plaques and prevents Aβ deposition in the brain with high selectivity for Aβ protofibrils.	Infusion-related reactions, headache, amyloid-related imaging abnormalities-edema (ARIA-E), ARIA-superficial siderosis, cerebral microhaemorrhages, ARIA cerebral microhaemorrhages, and falls.
Donanemab [51,77]	Monoclonal IgG1 antibody for earlier stages of AD reported by Eli Lilly.	Not available	This drug works by inducing microglial-mediated clearance of existing Aβ plaques with the intent of slowing the progressive decline in cognitive function associated with AD.	Gastrointestinal side effects including nausea, diarrhoea, and vomiting. Nervous side effects including cerebral microhaemorrhages and anxiety. Others including urinary tract infection and infusion-related reactions.

Abbreviations: ARIA (amyloid-related imaging abnormalities), ARIA-E (amyloid-related imaging abnormalities-edema), ARIA-H (amyloid-related imaging abnormalities-haemorrhage).

## 3. Potential Therapy to Target Neuroinflammation in AD

Neuroinflammation is considered to be a central factor in shaping neuronal vulnerability during AD pathogenesis, which has gained more focus recently. This includes an increase in the density of glial cells such as astrocytes and microglia in the brain and a change in their secretory profile from protective and anti-inflammatory to acute and pro-inflammatory. This triggers a cascade of changes at both the molecular level and at higher, macro levels which alters the healthy brain homeostasis to promote pathology. Among the effects is impaired Aβ processing, a harmful increase in the level of cytokines such as tumour necrosis factor (TNF)-α [78] and glutamate excitotoxicity due to failure to reuptake excess neurotransmitters from the synaptic cleft [79], causing dysfunction of neuronal networks and memory impairment [80]. Studies in both humans and rodents show that neuroinflammation is elevated in AD brain tissue compared to brain tissue from healthy, age-matched subjects (reviewed in [81]). Thus, accumulating evidence suggests that an inflammatory response, driven by activation of the brain’s innate immune cells, plays a crucial role in exacerbating neuronal damage and promoting disease progression [82]. Therefore, it is not surprising that new drug development and drug repurposing strategies aimed at targeting neuroinflammation have been considered in AD research. If these strategies were to work, it would be a revolutionary breakthrough in drug development as it would accelerate the drug development process, reduce the costs and risks inherent to drug development, and provide new therapeutic implications for clinically-approved drugs with different indications [83,84,85,86].

Here, we will briefly discuss these strategies and provide a novel perspective on targeting neuroinflammation.

### 3.1. Repurposing Established Antiviral and Anti-Inflammatory Drugs

Specifically, established antiviral and anti-inflammatory drugs are currently under scrutiny for their potential to modulate neuroinflammation in AD, which stems from recent evidence which suggests that pathogens such as viruses and bacteria are present in the AD brain [87]. Specifically, it has been proposed that the amyloid-β peptide, traditionally viewed as a hallmark of AD pathology, may function as an antimicrobial peptide within the framework of innate immunity. Antiviral drugs, such as Acyclovir and Penciclovir (Figure 1), traditionally used to target herpes simplex virus 1 (HSV-1), have shown promise in reducing both viral presence and amyloid-β accumulation in the brain [87,88,89]. Therefore, antiviral therapies may be considered as potential AD therapies.

Based on the central role of inflammation, anti-inflammatory drugs, mainly non-steroidal anti-inflammatory drugs (NSAIDs), are thought to have a protective effect against AD [90]. In one study, the use of ibuprofen, paracetamol, aspirin, and naproxen was linked to a significant reduction in AD compared to the non-pain-reliever group [91] (Figure 1). Recent systematic reviews and meta-analyses, including 18 observational studies and a randomized clinical trial (RCT), highlight that long-term NSAID use can lower the incident risk of AD by 28%, especially when used for durations longer than previously studied in trials like the Anti-inflammatory Prevention Trial, where 15 months of NSAID use proved to be insufficient [92,93]. These findings suggest that anti-inflammatory treatments, particularly long-term NSAID use, could potentially align with the neuroinflammatory hypothesis of AD, offering a promising avenue for reducing disease incidence and progression.

With the idea of neuroinflammation, the roles of resident glial cells such as microglia and astrocytes, along with endothelial cells and mast cells, are important in protecting the brain against foreign pathogens [94,95]. Microglia, the primary immune effector cells of the central nervous system (CNS), continuously survey the environment for potential threats, transitioning to an activated state characterised by cytosolic enlargement and the production of inflammatory cytokines and chemokines upon detecting invasive agents [96,97,98]. Similarly, astrocytes contribute significantly to mediating neuroinflammation and are integral in various neuroprotective functions, including maintaining the integrity of the blood–brain barrier (BBB) and buffering neurotransmitters [99]. Like microglia, astrocytes undergo morphological changes and increase their reactivity and secretion of cytokines and chemokines following injury, underlining their vital role in the neuroinflammatory response in AD [100]. This led us to propose a novel target, the endothelin B receptor (ET_B_R), that was related to astrocytes.

### 3.2. The Endothelin B (ET_B_R) Receptor as a Potential Therapeutic Target

The ET_B_R is one of the receptors in the endothelin family, which are predominantly expressed in the hippocampus and amygdala. Notably, astrocytes expressing ET_B_Rs showed high levels in ligand binding assays and studies of mRNA expression, highlighting its relevance in AD research [101,102]. Consequently, there has been an increasing focus on exploring the ET_B_R for its potential therapeutic applications [103,104,105]. Recent studies have revealed the crucial role of astrocytic ET_B_Rs in the development of traumatic brain injury (TBI) and its potential implications in the progression of AD [106]. Since the ET_B_R can activate astrocytes through the autocrine effect of its ligand endothelin-1 (ET-1), targeting ET_B_Rs is therefore considered as a precise way to regulate astrocyte activity.

Further research into the inhibitory effects of the ET_B_R on TBI models has provided evidence suggesting that the ET_B_R may be an effective target for controlling astrocyte activity during TBI [102,107]. This research also showed possibilities for understanding its implications in the progression of AD, making the ET_B_R a promising target in the field of neurodegenerative disease research [108,109].

A noteworthy advancement in this area is the use of the ET_B_R antagonist BQ788. Several studies have demonstrated its efficacy in reducing the number of reactive astrocytes, improving disruptions in the BBB, and decreasing brain swelling [102,110,111]. These outcomes indicate that antagonists of ET_B_Rs could play a crucial role in diminishing astrocyte activation, thereby offering relief in neurological disorders such as TBI and AD where astrocyte hyperactivation is a common factor.

The connection between reactive astrocytes in the pathogenesis of TBI and the successful inhibition of ET_B_Rs has laid a foundational basis for exploring the impact of ET_B_Rs in AD. While it remains to be clarified whether the mechanisms observed in TBI are directly applicable to AD, this research provides fresh and exciting perspectives on potential new therapeutic strategies in the treatment of AD.

## 4. Current Therapies for NPS in AD, Limitations, and Challenges

An under-researched aspect of dementia in general includes mood disorders such as anxiety and depression, among other NPS, that are often comorbid in patients with AD; a cross-sectional study assessing a cohort of 103 AD patients found that 51% presented with depressive symptoms—23% had major depression (MDD) and 28% were found to have dysthymia (a milder but longer-lasting form of depression) [112]. Patients tend to present with symptoms of anxiety at the mild cognitive impairment stage of AD. Anxiety is commonly described as a negative emotional state, which can be characterised by psychological symptoms (hypervigilance and feelings of worry and dread) and physiological changes (increased sympathetic tone, resulting in sweating and increased blood pressure and heart rate) [113]. In non-AD elderly patients, there is often a higher prevalence of anxiety and depression [114]; however, this prevalence is even higher in patients with AD, with over 70% displaying symptoms of anxiety, as observed in one study of 523 community-dwelling AD subjects [114], and it can be reasonably assumed that there is an association between AD and the development of mood disorders. A behavioural analysis study using transgenic mouse models of AD found that when subjected to a series of stressful stimuli, the mice exhibited anxiety-like defensive behaviours and risk assessment behaviours which can be equated to hypervigilant and avoidance behaviours in anxious humans [113].

However, there is currently little known about the direct relationship between AD and concomitant NPS and the cellular causative factors behind this correlation, yet the prevalence of comorbidity between AD and anxiety is highly apparent and there is an urgent clinical need to address this issue.

Currently prescribed medications for the treatment of anxiety and depression include SSRIs/SSNRIs and short-term benzodiazepines. Whilst the use of SSRIs is in accordance with the pathophysiology of depressive disorders (although major depressive disorders often require additional treatment options such as bupropion [115,116]), it may not be the most rational choice of drug design for anxiolytics.

The γ-aminobutyric acid (GABA) system is the main inhibitory system in the CNS that is involved in the pathophysiology of anxiety-related disorders. Synaptic and extrasynaptic γ-aminobutyric acid type A receptors (GABA_A_Rs) have different subtypes and each subtype is composed of different subunits which are encoded by different genes and determine the pharmacological properties of the GABA_A_Rs [117,118]. Subunit combinations and synaptic GABA_A_R activation cause phasic inhibition and most of them consist of α1, β2, and γ2 subunits. Benzodiazepines can only modulate GABA_A_ receptors containing the γ subunits which are expressed synaptically; when bound to these subunits, they can result in immediate and effective relief of symptoms related to anxiety. However, their use is associated with concomitant adverse drug reactions (ADRs) and sedative, amnestic, or anticonvulsant effects as well as impaired coordination and visual disturbances, especially amongst the elderly who are particularly susceptible due to changes in pharmacokinetic and pharmacodynamic profiles. A further unpleasant side effect profile has deemed benzodiazepines as unsafe for long-term use due to addiction and dependence problems. Therefore, SSRIs are currently considered the first-line therapy for the management of mood-related disorders even though this class of drugs may not be in accordance with the primary pathophysiology of anxiety. This highlights the need for the development of more rationally designed medications that can target the main pathophysiological pathways required for the management of anxiety disorders.

## 5. Insights into More Targeted Therapy for NPS in AD

Experimental evidence suggests that cognitive impairment and emotional dysregulation in AD are linked to synaptic excitation–inhibition imbalance in the brain caused by damage to the hippocampus, prefrontal cortex (PFC), and limbic subcortical regions [113].

The role of the major inhibitory neurotransmitter GABAergic system in the brain beyond the use of benzodiazepines is often overlooked, particularly the extrasynaptic components that could offer an alternative solution to the problems associated with benzodiazepines. This view stems from the evidence that GABA_A_R signalling deficits have been linked with many neurological disorders such as anxiety, depression, AD, chronic alcohol dependence, and schizophrenia [119,120], which suggests there is a clinical need for subtype-specific compounds that target the GABA_A_R.

Although some studies suggest that GABAergic neurons are more resistant to neurodegeneration in AD [121,122] relative to cholinergic and glutamatergic neurons, others have reported that aberrant brain GABA levels are present in AD and mood disorders as well as in normal ageing [123,124,125,126]. Furthermore, evidence suggests a selective vulnerability of inhibitory interneurons during the disease progression of AD [127,128]. Therefore, there is scope for the development of drugs that can preferentially bind to specific subunits of the GABA_A_R family that will have fewer ADRs than the current benzodiazepines used clinically. One area of focus for the future should be to consider the extrasynaptic components of the GABAergic system that are exclusively located on extrasynaptic terminals.

Activation of extrasynaptic GABA_A_Rs leads to tonic inhibition, indicating that a low concentration of GABA is required for their activation. The tonic inhibition of GABA_A_Rs leads to a stronger inhibitory effect on excitatory neurons without affecting their sensitivity, which can reduce the risk of resistance [129]. The main subunits of interest in the extrasynaptic GABA_A_Rs are the δ subunit-containing receptors, which are mostly coupled with α4 subunits. These receptors have high GABA sensitivity with slow de-sensitivity [130,131]. The αβδ and αβγ GABARs have different pharmacological properties, and GABA is considered a partial agonist of αβδ GABARs.

Another avenue for potential new therapies for general NPS is focusing on drug development centred around neuroactive steroids (NASs), which act as positive allosteric modulators (PAMs) of both synaptically- and extrasynaptically-expressed GABA_A_Rs. NASs also lead to increased expression of extrasynaptic GABA_A_Rs through a metabotropic mechanism [132]. There are also extrasynaptic GABA_A_R-selective drugs such as Gaboxadol (THIP), which acts on α4β3δ subunits, that can potentially be used for the treatment of anxiety and depression. Gaboxadol selectively targets the extrasynaptic GABA_A_Rs that contain α4 and δ subunits. It has been shown that Gaboxadol has anti-anxiolytic (as well as sleep-inducing) effects but due to its side effect profile and inconsistent therapeutic benefit, its use was discontinued. However, due to its selectivity in targeting extrasynaptic GABA_A_Rs, which regulate tonic inhibition, it remains an investigational therapy of interest. Furthermore, it has been proven that there is an altered expression of δ subunit-containing GABA_A_Rs in conditions such as anxiety and depression, which further suggests that targeting extrasynaptic GABA_A_Rs can be useful in the management of mood disorders. Currently, other than Gaboxadol, such subunit-specific therapies have not been identified. However, there are neurosteroid modulators such as Brexanolone (SAGE-547), Zuranolone (SAGE-217), and Ganaxolone that can target both synaptic and extrasynaptic GABA_A_Rs [133,134,135,136].

An analysis of the clinical trials with Zuranolone, Ganaxolone, and Brexanolone indicated that these neurosteroids were effective in the management of depressive-related disorders. The included studies had limitations, including small sample size and unknown long-term effects of the therapies. These studies were placebo controlled. Further studies are warranted where the effectiveness of these therapies is compared with the standard SSRIs/SNRIs to identify which therapy is more effective. One of the main advantages of these neurosteroid medications is their fast onset of action in comparison to SSRIs, where the effects are commonly observed after up to 12 weeks of use. Results of an indirect trial comparison (ITC) that compared the HAM-D (Hamilton Depression) rating score levels of depression of trials with Brexanolone in post-partum depression with SSRIs indicated that Brexanolone showed a larger change from baseline in HAM-D scores. This, combined with the rapid response rate of Brexanolone, indicates that this therapy can be advantageous over SSRIs, particularly in conditions such as post-partum depression where a rapid response is crucial for the patient as well as the child [24]. Direct comparison studies are needed to investigate this hypothesis further.

Pregnenolone used for the management of bipolar depression did not indicate a notable improvement in the HAM-D score. For the primary measurements, there was a notable increase in the remission rate as indicated by the IDS-SR (Inventory of Depressive Symptomatology Self Report) score; however, this may be due to a lower baseline IDS-SR score in the pregnenolone group. The use of pregnenolone was also associated with an improvement in the HARS (Hamilton Anxiety Rating Scale). Figure 2, illustrates a summary of our discussion points in this review, summarizing the current treatments available for alleviating cognitive decline and NPS associated with AD, and potential targets for future drug design. 

## 6. Conclusions

The proportion of the ageing population in the world is increasing; therefore, so is the incidence of AD. We need better medications for cognitive decline as well as other symptoms of AD. This problem is accentuated by the fact that treatments do not yet provide a cure for AD nor a rationally designed therapy to manage symptoms of anxiety or agitation that are commonly associated with the disease. Consequently, the call for developing more suitable and effective pharmaceutical treatments for AD and for the management of associated anxiety-related disorders is all the more urgent as there is currently a lack of selective and rationally designed medications that can prevent, halt, or reverse disease progression. This review advocates for the exploration of novel therapeutic avenues, including the selective targeting of specific pathways such as the ET_B_R system to reduce neuroinflammation and GABA_A_Rs to mitigate hyperactivity associated with cognitive deficits. By focusing on these unexplored angles, we aim to develop treatments that are not only more effective but also come with fewer side effects, providing a promising direction for future AD management strategies.

## Figures and Tables

**Figure 1 ijms-25-05169-f001:**
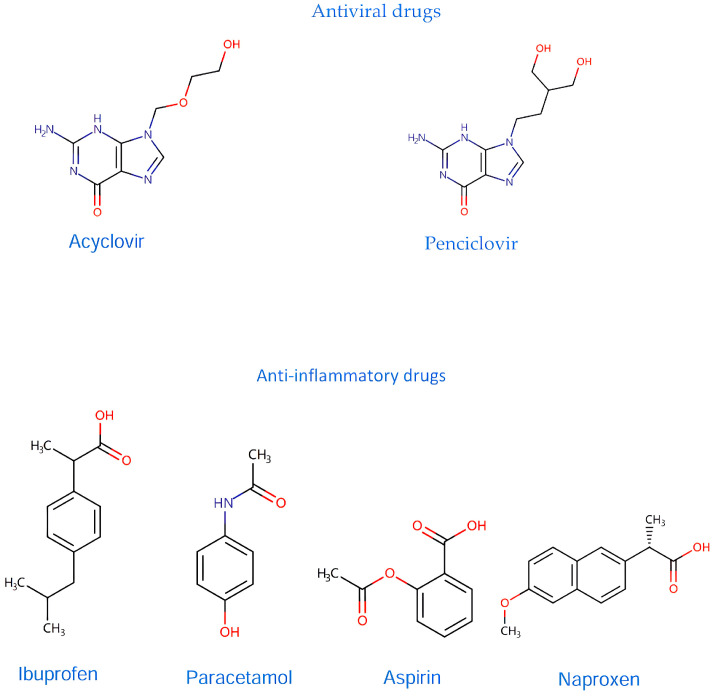
All the chemical structures in antiviral and anti-inflammatory drugs.

**Figure 2 ijms-25-05169-f002:**
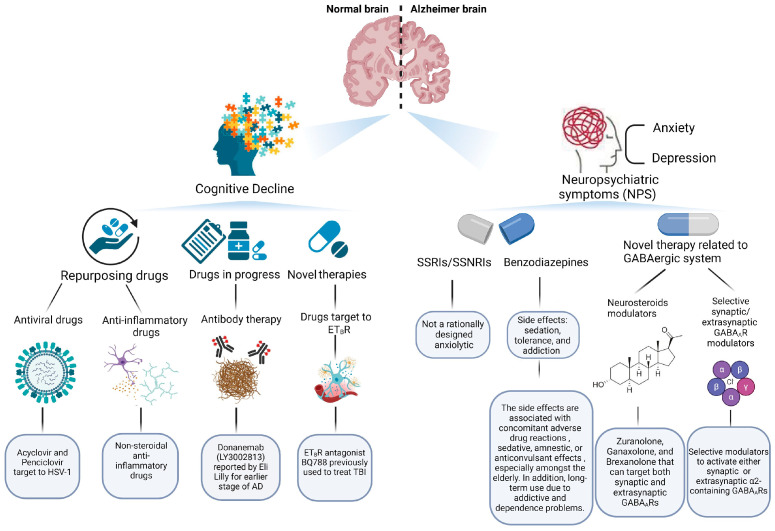
The summary diagram of the graphical representation of the current and potential future treatments for alleviating cognitive decline and NPS.

**Table 2 ijms-25-05169-t002:** Clinical trials for AD therapy from 2024.

Treatment of Drugs	Phase	Ages for Study	AD Stages	Treatment Purpose
AL002 and placebo	Phase 2	50–85	Early	Efficacy and safety
SHR-1707 and SHR-1707 placebo	Phase 2	50–85	Mild	Safety and pharmacodynamics
ALZ-801	Phase 3	50–85	Early	Long-term safety and efficacy
ADEL-Y01 and placebo	Phase 1	18–80	Mild	Safety, tolerability, PK, and PD
ACU193	Phase 2 and Phase 3	50–90	Early	Efficacy and safety
Benfotiamine and placebo	Phase 2	50–89	Early	Safety, effectiveness, and tolerability
Masitinib and placebo	Phase 3	≥50	Mild to moderate	As an adjunct to a cholinesterase inhibitor
DDN-A-0101 and placebo	Phase 1	19–75	Early	Safety, tolerability, and pharmacokinetics
BMS-986446 and placebo	Phase 2	50–80	Early	Efficacy, safety, and tolerability
Daridorexant and placebo	Phase 4	60–85	Mild to moderate	Efficacy and safety

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
