# Peer review of "Advances in Therapeutics to Alleviate Cognitive Decline and Neuropsychiatric Symptoms of Alzheimer’s Disease"

_ijms, 2024, doi:10.3390/ijms25105169_

Round 1

Reviewer 1 Report

Comments and Suggestions for Authors

1.       The authors summarized the current FDA-approved treatments for AD in Table 1. Please present the chemical structure of those drugs.

2.       In section 3.1, it would be better to provide examples (with their chemical structures) of the treatments.

3.       In section 3.2, references are missing in the second paragraph.

4.       What is GABAAR in line 304? Is it different from GABA receptor?

5.       In conclusions, future direction for development of AD treatments should be provided.

6.       The references are not in correct format. It should be fixed.

7.       In abstract, what is “(NICE, 2021)”? It should be deleted.

Author Response

Comments and Suggestions for Authors

Thank you for giving us the opportunity to revise our manuscript.

We appreciate the support and detailed comments suggested by the reviewers.

Our response and corrections are in blue in the manuscript.

  1. The authors summarized the current FDA-approved treatments for AD in Table 1. Please present the chemical structure of those drugs.

Response: We apologise for not including the chemical structures, which have now been added to Table 1. Page 5-6. Thank you for this suggestion.

  1. In section 3.1, it would be better to provide examples (with their chemical structures) of the treatments.

Response: We appreciate the reviewer’s suggestion. The chemical structures of these molecules, now shown in Figure 1 Page 8. The examples of anti- inflammatory drugs have been mentioned in line 244.

  1. In section 3.2, references are missing in the second paragraph.

Response: Apologies for the missing information, we have included the missing reference.

  1. What is GABAAR in line 304? Is it different from GABA receptor?

Response: GABAARs has been defined better now on page 9, line 343 as γ-aminobutyric acid type A receptors.

  1. In conclusions, future direction for development of AD treatments should be provided.

Response: Thank you for this suggestion, we have included the future direction on page11, line 435-440:

“This review advocates for the exploration of novel therapeutic avenues, including the selective targeting of specific pathways such as the ETBR system to reduce neuroinflammation and GABAARs to mitigate hyperactivity associated with cognitive deficits. By focusing on these unexplored angles, we aim to develop treatments that are not only more effective but also come with fewer side effects, providing a promising direction for future AD management strategies.”

  1. The references are not in correct format. It should be fixed.

Response: The template uploaded on the journal webpage was approved and the references style inserted were from the ijms office. We have double checked the references format and it has been fixed.

  1. In abstract, what is “(NICE, 2021)”? It should be deleted.

Response: This has been deleted.

Reviewer 2 Report

Comments and Suggestions for Authors

The manuscript “Advances in therapeutics to alleviate cognitive decline and neuropsychiatric symptoms of Alzheimer’s Disease” addresses current and novel targets for AD therapy. The review is interesting, but I have some suggestions to improve the manuscript:

1)      Proofread the text. The reference in line 9 of the abstract should be removed and, in line 76, the reference is not properly formatted. In lines 112 and 113, a different font was used.
In figure 1, please correct “anitibody” to “antibody” and “noval” to “novel”. What is the meaning of the sentence “The main novel aspects that hadn’t been illustrated”?

2)      Although the authors opted to discuss different therapeutic options that provide originality to the manuscript, I recommend briefly referring to new tau therapies such as tau antisense oligonucleotides and anti-tau oligomers antibodies.

3)      I also suggest searching in the clinical trials gov database and indicating the clinical trials for AD therapy, ongoing for 2024.

4)      Aducanumab is being discontinued by Biogen. This information is also worth mentioning.

5)      In table 1, indicate the side effects of donanemab.

Comments on the Quality of English Language

Please proofread the manuscript, minor alterations are required.

Author Response

Ref 2:

  • Proofread the text. The reference in line 9 of the abstract should be removed and, in line 76, the reference is not properly formatted. In lines 112 and 113, a different font was used.
    In figure 1, please correct “anitibody” to “antibody” and “noval” to “novel”. What is the meaning of the sentence “The main novel aspects that hadn’t been illustrated”?

Response: Apologies for the mistakes. We have proofread the text and fixed all the reference. As the sentence “The main novel aspects that hadn’t been illustrated” doesn’t make sense, we deleted that. All the spelling errors that appeared in Figure 2 have been corrected, sorry for the mistakes.

  • Although the authors opted to discuss different therapeutic options that provide originality to the manuscript, I recommend briefly referring to new tau therapies such as tau antisense oligonucleotides and anti-tau oligomers antibodies.

Response: Thanks for the suggestions. I have included the following on  Page 4 line170-189:

“In addition to therapies targeting Aβ, several treatment strategies focus on tau pathology, including tau antisense oligonucleotides and anti-tau oligomer antibodies. Increasing evidence supports the role of hyperphosphorylated tau aggregation as a central contributor to neurodegeneration in AD [56-58]. The tau protein, primarily expressed in neurons, is encoded by the microtubule-associated protein tau (MAPT) gene [59, 60]. Preclinical studies suggest that reducing tau can prevent certain deficits mediated by amyloid-beta (Aβ), underscoring tau's pivotal role in Aβ toxicity during the early stages of AD pathogenesis [61, 62]. At the close of 2023, Biogen revealed new Phase 1b clinical data for BIIB080, an investigational antisense oligonucleotide (ASO) therapy targeting tau in patients with mild AD [62]. BIIB080 is engineered to target MAPT mRNA to lower tau protein production. Inhibiting MAPT expression to reduce tau levels is a crucial strategy, which is a key mechanism of disease effect directly targeting AD patients [61]. While BIIB080 targets the reduction of tau protein production by inhibiting MAPT mRNA, another promising strategy involves the anti-tau oligomer antibody APNmAb005 [63-65]. APNmAb005 is a humanized monoclonal antibody designed to block the synaptic toxicity caused by tau oligomers[65]. APNmAb005 selectively binds to tau oligomers and aggregates, primarily within the synapses of pathological brain tissues, effectively inhibiting tau propagation[64, 65]. Currently, APNmAb005 is undergoing Phase 1 clinical trials to evaluate its safety and tolerability, representing a novel approach in tau immunotherapy for neurodegenerative disorders[64].

In 2024, some clinical trials are actively exploring therapies for AD. A significant trial involves AL002, targeting early AD stages and assessing changes in cognitive and biomarker outcomes over up to 96 weeks. More information is mentioned in Table 2 [66]. These trials reflect a broad effort to target various aspects of AD pathology and progression.”

  • I also suggest searching in the clinical trials gov database and indicating the clinical trials for AD therapy, ongoing for 2024.

Response: Thanks for the suggestions. I have mentioned the information on Page 4 line190-194 and added Table2 to show more clinical trials.

4)      Aducanumab is being discontinued by Biogen. This information is also worth mentioning.

 Response: Thanks for the suggestion. We have added the information of ‘Aducanumab is being discontinued by Biogen’ on page3 line 136-140.

5)      In table 1, indicate the side effects of donanemab.

Response: Thanks for the suggestions. We have added the side effects of donanemab in Table 1.
